# Anthropogenic Influences on Distance Traveled and Vigilance Behavior and Stress-Related Endocrine Correlates in Free-Roaming Giraffes

**DOI:** 10.3390/ani11051239

**Published:** 2021-04-25

**Authors:** Ciska P. J. Scheijen, Sean van der Merwe, Andre Ganswindt, Francois Deacon

**Affiliations:** 1Department of Animal, Wildlife and Grassland Sciences, University of the Free State, Bloemfontein 9301, South Africa; DeaconF@ufs.ac.za; 2Rockwood Conservation Fund NPC, Griekwastad 8365, South Africa; 3Department of Mathematical Statistics and Actuarial Science, University of the Free State, Bloemfontein 9301, South Africa; vanderMerweS@ufs.ac.za; 4Mammal Research Institute, Department of Zoology and Entomology, University of Pretoria, Hatfield 0028, South Africa; andre.ganswindt@up.ac.za; 5Center for Veterinary Wildlife Studies, University of Pretoria, Onderstepoort 0110, South Africa

**Keywords:** habituation, anthropogenic influences, fecal glucocorticoid metabolites, stress, vigilance, distance traveled

## Abstract

**Simple Summary:**

Change in an animal’s behavior due to anthropogenic influences is often expressed in a change in movement patterns and increased vigilance and can result in the secretion of stress-related hormones. However, animals can get habituated to human presence after repeated stimulation. We aimed to obtain a first insight into the effect of human observers on the behavior and stress-related hormone concentrations of free-roaming giraffes as well as their habituation process. Giraffes walked further distances and had elevated fecal glucocorticoid metabolite (fGCM) concentration (stress hormone) in the presence of humans, but anthropogenic influences on their distance walked and fGCM concentration decreased with the increase of habituation. The giraffes were vigilant towards human observers; however, the percentage of time spent on observing a human observer did not decrease with the increase of habituation.

**Abstract:**

Giraffes are an important tourist attraction, and human presence to wildlife is increasing. This has an impact on an animal’s behavior and its endocrine correlates. Studies on other species show alterations in movement patterns, vigilance, and stress-related hormone levels in the presence of humans. Limited information is available on how anthropogenic activities alter giraffe’s behavior, social structure, and related endocrine parameters. The purpose of this study was to obtain insight into anthropogenic influences on giraffe’s behavior and adrenal activity. We used GPS devices mounted onto giraffes to compare the distance walked in the presence or absence of human observers. We also conducted behavioral observations to assess their vigilance and collected fecal samples to analyze their fecal glucocorticoid metabolite (fGCM) concentrations. Giraffes walked significantly further distances in the presence of humans, but the cumulative time that observers were present decreased the hourly distance walked with an observer present, suggesting that the giraffes were becoming habituated. The number of observers present significantly increased the percentage of time spent on observing an observer as well as the number of unhabituated individuals present in the herd. The percentage of time spent observing a human observer did not decrease with the increase of habituation. Last, fGCM concentrations increased with human presence but decreased when individuals became habituated to human presence. More research is needed to understand the effect of anthropogenic influences in different scenarios (e.g., tourism, vehicles, hunting, etc.).

## 1. Introduction

Behavioral ecology studies where observers are present often rely on habituated focal individuals or groups [1]. Habituation is a behavioral response resulting from repeated stimulation that does not involve sensory adaptation [2] and can take place as a result of the repeated intended presence of humans, e.g., for research purposes [1] or in areas where human presence is common, e.g., in relation to tourism, human settlements, etc. [3]. Habituation usually requires a prolonged period of repeated stimulation to have any sort of effect [4], and there may be individual differences in the process of habituation, depending on the life history and the character of the individual [5,6].

Habituation can cause changes to an animal’s behavioral patterns and physiological status [7]. Therefore, human presence can result in atypical behavioral alterations, as animals can see humans as predators [8] or can be naturally elusive [4]. Human presence can change the vigilance behavior and potentially the social structure of animals who live in groups [3,9]. For example, wildlife viewing of polar bears (*Ursus maritimus*) in Canada influence the bear’s vigilance behavior [10], and human observers and settlements influence samango monkey’s (*Cercopithecus mitis erythrarcus*) [1] and giraffe’s (*Giraffa camelopardalis*) [11] movement patterns and prey–predator relationships. 

Once habituated, it is often assumed that an animal’s behavior is relatively independent of human presence [12]. It is however known that humans can influence animal behavior, even after habituation [1,13]. From the literature, it is known that human presence to habituated samango monkeys can cause changes in movement patterns of the species and influence prey–predator relationships [1]. Canine (1990) [12] demonstrated that habituated tamarins (*Saguinus labiatus*) remained in their nest box for longer periods of time with observers present. If focal animals adapt their behavior when humans are present, then behavioral studies on their “normal” daily activity with observers present would be biased [13].

It can be challenging to assess changes in animals’ behavior caused by human presence [1,9,13]. GPS devices mounted onto individuals and methods such as “giving-up density” (GUD) studies [1] to observe behaviors with and without human presence have been used in a variety of species.

Theunerkauf et al. (2001) [5] showed that all radio-tracked activities of wolves (*Canis lupus*) were correlated with human activity, however, there was individual difference in the strength of these correlations. In contrast, radio-collared and habituated capuchin monkeys did not show a difference in movement patterns depending on observers’ presence [14], indicating species-specific variation in response to observers as well. Even though one can gather valuable information with GPS devices, such devices are associated with high costs [15]. In addition, it can be a challenge to get approval and permits to mount such devices on animals, especially when endangered species are involved. A cheaper and non-invasive alternative could be the use of the GUD method. Using this method, Nowak and colleagues (2014) [1] demonstrated that two groups of habituated samango monkeys foraged more often at lower levels with humans present compared to when humans were not present, suggesting that human observers possibly lowered their perceived risk of fear for terrestrial predators.

The habituation process can also cause changes to the physiological status of the animals [7], for example, changes in nervous system output including reflexes towards stimuli such as sweating and muscle contractions [2]. The disruption of homeostasis in an animal due to environmental perturbation usually leads to an the of the hypothalamic–pituitary–adrenal (HPA) axis, resulting in an increase of glucocorticoid output [16]. However, a study by Pitman et al. (1990) [17] showed that rats became habituated to the repeated exposure to a particular stressor, to a degree at which habituation was negatively related to the stress stimulus intensity. Further, Jack et al. (2008) [18] showed that unhabituated white-faced capuchins (*Cebus imitator*) had elevated fecal glucocorticoid metabolite (fGCM) concentrations in the presences of humans, but the concentration decreased with the duration of habituation. Similarly, a study by Shutt et al. (2014) [19] showed that Western lowland gorillas (*Gorilla gorilla*) demonstrated lower fGCM concentrations after long-term habituation compared to animals for which habituation was still ongoing. Nevertheless, the overall fGCM concentrations in animals of both groups were higher compared to the hormone metabolite concentrations in animals in the absence of human observers, suggesting that habituation reduced fGCM concentration in gorillas in the presence of humans, but this concentration remained elevated even after habituation. 

Most wildlife in South Africa live in constrained and fenced environments [20], such as game reserves, national parks, and private game farms, where often human activities (like game viewing) take place. Giraffes (*Giraffa camelopardalis*) are an important tourist attraction [21], and increasing research is being conducted on giraffe behavior due to their classification as “vulnerable” in the IUCN red list since 2016 [22]. However, information on physiological correlates of giraffe behavior in the presence of humans is limited. Further, a better understanding of potential observer’s effects on giraffe behavior and related physiological status can contribute to decision making in conservation strategies. 

Male giraffe reproductive tactics involve a trade-off between food intake and breeding opportunities, as they roam between female herds in search of a female on heat, which is an activity with a high energy cost [23,24,25]. The female reproductive system also demands high energy, as females fall pregnant while lactating [26]. The influence of human presence on the time spent moving (increasing energy cost and lowering energy intake as well as time for social interactions) and on the percentage of time that a giraffe spent on observing an observer (lowering energy intake and time for social interactions) was investigated in this study. 

Prolonged stress can have an impact on an individual’s health and survival by affecting the immune [16,27,28] and reproductive functions [28,29,30] as well as disease resistance of the stressed individual [16,19,28]. Therefore, the effect of human presence on fGCM concentration was also investigated in this study. 

Hourly distances traveled, behavioral observations, and fGCM concentration for the focal individuals in the presence or absence of human observers were compared. 

It was expected that giraffes would walk further distances when humans were present compared to when humans were not present, but over time this response would be reduced. Secondly, we expected that giraffes would “observe observers” (OO) when present, and the percentage of time spent on OO would reduce over time. Finally, it was expected that the presence of observers elicited an fGCM response in giraffes, but habituation would reduce this response over time.

## 2. Materials and Methods

### 2.1. Study Area and Population Demography

The study was carried out at Rooipoort Nature Reserve (RNR) (28°36’59” S, 24°15’28” E), South Africa (Figure 1), located in the transition zone of the Karoo, Kalahari, and grassland [31]. The total reserve (42,647 ha) is split by a fence into two parts, with giraffes roaming freely in the largest part (34,500 ha). The climate consists of dry and cold winters and wet hot summers [31]. Limited evergreen foliage is available all year round, although in the dry season (May until October) there is a phenological effect with deciduous foliage resulting in less food available compared to the wet season (November until April). Besides salt licks, no additional feeding supplements are provided. RNR has 12 artificial permanent waterholes, and during the wet season, there can be numerous temporary and sporadic natural drinking locations. Besides giraffes, the reserve contains other ungulates and no large predators, except for brown hyena (*Hyaena brunnea*), which are mainly scavengers. A weather station (Davis Vantage pro2^®^, Davis Industrial Ptl (Ltd), Hayward, CA, USA) was installed at the main camp, collecting hourly basic weather data (rainfall, wind speed, wind direction, humidity, and temperature). 

Six adult males, 26 adult females, 7 sub-adults, 1 juvenile, and 13 calves were identified at RNR during the time of observation. All giraffes were born at RNR, and no new introductions of giraffes to RNR occurred since 1974. None of the giraffes or any of the other wildlife species were habituated to human presence at the start of the study. In fact, up to the start of the study, hunting (by foot) still occurred at RNR, which included hunting of giraffes.

### 2.2. Study Animals

In October 2017, 18 giraffes (all 6 adult males, 1 sub-adult male, and 11 adult females) were equipped with GPS devices, designed, and created by Africa Wildlife Tracking (AWT, Pretoria, South Africa), following procedures described by Deacon and Smit (2017) [32]. Although the devices have been successfully used before by Deacon (2015) [33], due to some technical and logistical complications with some of the devices or fittings, detailed and prolonged data were only gathered for four adult males and three adult females. The locations of these seven giraffes were logged hourly, and distances between two following locations were calculated using QGIS Desktop 3.6.0. Occasionally, the devices did not find satellite signal when a location recording was required, and therefore no location was logged for that specific hour. Distance data that did not have a valid logged reading were excluded from the analysis. Locations of giraffes were logged for between 5 and 12 months, depending on the individual (Table 1), with a total of NLogs = 33,772. 

### 2.3. Experimental Approach and Study Design

The criteria for fitting GPS devices to the giraffes were good health and body conditions, confirmed by two experienced wildlife veterinarians at the time of fitting the collars. For safety reasons, captures were only done in the central and eastern parts of the reserve. The western part consists of mountainous and rocky environments causing difficulties for the capture team to reach it. Once the animal was darted with an immobilizing drug called Thianil^®^ (Thiafentanil oxalate 10 mg/mL; wildlife pharmaceuticals Ply. (Ltd.); White River, South Africa), it was important to reach the giraffe on the ground as soon as possible. Because individuals were captured in specific areas, they were more likely to share habitats from time to time (although the entire reserve was free to be roamed by any individual during any given time). The giraffes inhabiting the western part were not fitted with GPS collars, were not seen often, and were assumed to be still unhabituated by the end of the study. There was an active diamond mine in the southwestern part of RNR, and observers were prohibited from following the giraffes if they entered the mining area. The GPS devices revealed that only the males visited the mine area from time to time, none of the females who were fitted with GPS devices ever went to the mining area. However, there are unknown females with calves inhabiting that area.

To our knowledge, defining what is considered a habituated wild animal has only been published for primates [34]. With little to no literature or information available with specific reference to the hours needed to habituate giraffes, a scoring criterion on what is considered a habituated or unhabituated animal was created, using our own data as an example. Individuals were considered to be “unhabituated” when they were seen on less than 25 days throughout the study period (unhabituated individuals were seen a median of 6.5 times; habituated (at the end) individuals, a median of 49 times). The times “seen” involves any time observers encountered the giraffe, so it is possible observers were only present for less than a few hours. 

Due to the shorter lifetime of the GPS device fitted to one female, F7 (Table 1), we report fewer behavioral observer hours for her compared to the others. Behavioral observations were done for a total of 1861 h (mean = 266 h, min = 150 h, max = 339 h). Human observers were present for the focal individuals for a total of 2328 h (mean = 333 h, min = 188 h, max = 395 h) over 12 months. The latter value includes any presence of humans, e.g., during behavioral observations, when the individual was present in the herd while observers were collecting behavioral data of another individual, as well as when observers were only collecting fecal samples.

### 2.4. Behavioral Observations

The focal individuals’ activity was recorded continuously from dawn to dusk on observation days (continuous focal animal sampling) [35], while the group composition and size were also recorded. Observers used a vehicle to get to the nearest location point of the focal individual, then started following the focal individual by foot for behavioral observations. Animals were identified by their spot patterns. For behavioral observations, a modified ethogram by Seeber et al. (2012) [36] was used (see supplemental materials for the modified ethogram). A change of behavior was considered if the focal individual performed a different activity for more than 15 s. It was also possible that a giraffe performed two behaviors at the same time (for example, ruminating and locomotion or ruminating and “observing an observer”); in those cases, both behaviors were recorded. 

For this study, only the time a giraffe spent on ‘observing an observer’ (OO) and ‘out of sight’ was recorded. OO was recorded when the focal individual was actively watching a human observer.

### 2.5. Fecal Sample Collection and Analysis

A total of 159 fecal samples were collected, of which 80 had no observers present, and 79 had at least one observer present (N samples per individual: F2 = 28, F6 = 19, F7 = 23, M1 = 24, M4 = 23, M5 = 20, M7 = 22). Samples were collected within 30 min of defecation following standardized procedures [37], were immediately stored on ice in the field, and frozen within six hours. Samples remained frozen until further processing at the Endocrine Research Laboratory, University of Pretoria, South Africa. Fecal steroids were extracted following established protocols [37,38] and subsequently analyzed for fGCM concentration. There is a lag time of one to two days between circulating hormone concentrations being reflected in faeces [38]. Therefore, samples collected following a day without humans present were treated as “no observers present”. Immunoreactive fGCM concentrations were determined using an enzyme-immunoassay (EIA) previously established for giraffe [37,38]. Assay characteristics including antibody cross-reactivities are provided by Möstl et al. (2002) [39]. The sensitivity of the assay at 90% binding was 1.2 ng/g fecal dry weight (DW). Inter-assay coefficients of variation (CV), determined by repeated measurements of low- and high-quality controls, were 10.39% and 11.75%, respectively, and intra- assay CV were 5.65% and 6.11%, respectively. 

### 2.6. Data Analysis

Statistical analysis was done using the R software package (https://www.R-project.org 01/01/2021). A linear mixed model [40] was used to test the human observers’ impact on the hourly distance walked by giraffes. As the distribution of distances walked was skew to the right, hourly distance data were log-transformed. The regression coefficients are reported as percentage change for easier interpretation. Because animals have innately different properties and behaviors, the individuals were modeled as a random set from a larger sample (with each animal having its unique properties). Because males tend to walk greater distances compared to females, sex was set as a fixed effect. Giraffes are diurnal animals and therefore more likely to be more active during the day; human observers were also only present during daytime, hence, “time of the day” was controlled for in four categories: (1) 00:00–6:00, (2) 6:00–12:00, (3) 12:00–18:00, and (4) 18:00–00:00. To control for seasonal differences in distances walked (due to water and food availability), season (dry/wet) was added as a fixed effect. The factors of interest (human presence and cumulative time with observers present) were both added as fixed effects on their own and random effects by individual to the analysis, in order to separate the general impact from the effects on specific individuals. To test the effect of the number of unhabituated individuals present on the distance walked by the focal individuals, with observers present, a separate linear mixed model was used. In this model, only the hours when human observers were present were included, as none of the unhabituated individuals were equipped with GPS devices and, therefore, there we had records of their presence to the focal individuals without human observers present. This model controlled for cumulative observation time, sex, season, and group size as fixed effects, as well as individual as a random effect.

A linear mixed model was used to assess what factors best describe the percentage of time a target subject spent observing the observer (OO). Individual was added as a random factor to the model. As it is known for many species that herd size reduces the vigilance of an individual [41], herd size was added to the model as a fixed factor. The factors of interest that were included as fixed factors were: (1) number of observers, (2) number of unhabituated individuals, (3) the time observers were present on a single day, and (4) the cumulative time observers were present for an individual prior to that point.

To investigate correlations to fGCM concentrations, a linear mixed model was fitted. Again, the individual was added as a random factor. It is known that fGCM concentrations can differ between males and females [16]; thus sex was added as a fixed factor. The final model included only these factors of interest: human observers present, the cumulative time of observer present, and sex, added as fixed factors, as well as individual as a random factor. Possible control variables, such as distance walked, season, rain, and temperature were found to be insignificant in the presence of the factors of interest, and including them in the model resulted in higher Akaike Information Criterion (AIC) values; thus, we report the model without those control variables. 

## 3. Results

### 3.1. Effect of Human Presence on Distance Traveled

Male giraffes significantly walked further distances compared to females (*p* = 0.001), with an overall average of 44.1% added when controlling for individual differences, time of the day, season, human presence, and cumulative time of observers present. The average distances per hour were 275.3 m for females and 404.5 m for males (Table 2). The time of day had a strong effect on the distance walked (*p* = 0.000) (Figure 2). Using 00:00 to 06:00 as a baseline, giraffes walked 237.6% further in the morning (06:00 and 12:00), 231.3% further in the afternoon (12:00 and 18:00), but 16.5% less at night (18:00 and 24:00).

During the dry season, the giraffes walked overall 18.9% further (*p* = 0.000) compared to the wet season (Figure 3). The presence of an observer significantly (*p* = 0.000) increased the hourly distance that a giraffe walked (Figure 3) by 66.6% (95% confidence interval 50.3% to 84.7%). The cumulative time that observers were present decreased the hourly distance walked by the giraffes with an observer present (*p* = 0.011) to the extent that it would take an average of 442 h of human presence to habituate a giraffe if circumstances and time frames are similar to this case study (Figure 4). The number of unhabituated individuals present in the herd significantly (*p* = 0.002) increased the distance walked on a day when human observers were present. The distance increased by 13.2% per extra unhabituated individual (Figure 5).

### 3.2. Effect of Human Presence on Vigilance Behavior

A larger herd size decreased the percentage of time spent OO significantly (*p* = 0.001), and there was a clear difference between the seven focal individuals in terms of their general tendency in time spent OO (average time spent OO per observation session; F2 = 6.8%, F6 = 8.81%, F7 = 4.72%, M1 = 5.63%, M4 = 2.15%, M5 = 9.88% and M7 = 6.2%). The number of observers present increased the percentage of time spent OO significantly (*p* = 0.001). In addition, the number of unhabituated giraffes present in the herd increased the percentage time spent OO of the focal (habituated) individual significantly (*p* = 0.005) (Figure 6). The time an observer was present on a single day did not significantly influence the percentage of time spent OO, and neither did the cumulative time presence. The variables were, nonetheless, kept in the model because they lowered the AIC value and were important controls for the other factors of interest. 

### 3.3. Effect of Human Presence on fGCM Concentration

There was no evidence in our dataset that fGCM concentration is affected by distance traveled or other environmental factors considered; therefore, these variables were excluded from the model. Overall, male giraffes had 45.8% lower fGCM concentrations compared to the opposite sex (*p* = 0.014). Further, there was a tendency for giraffes to have higher fGCM concentrations when an observer was present (*p* = 0.063) (Figure 7). Over time, the focal animals expressed lower fGCM concentrations in relation to the cumulative time human observers were present (*p* = 0.007) (Figure 4). Our model suggests an expected time to habituation of 427 h, indicating roughly zero human impact on fGCM concentrations, which is in line with what was suggested by the distance walked.

## 4. Discussion

Human presence can alter wildlife species’ movement patterns but also pose a risk to their survival and reproduction success [34]. In Kellerwald-Edersee National Park in Germany, off-trail hiking induced flight of the red deer (*Cervus elaphus*), and the animals avoided trails during the day but not at nighttime when hikers are absent [42]. In line with these findings, the results of this study suggest that giraffes were avoiding human observers, as the hourly distance walked by giraffes increased with human presence. Towards the end of the 12 months, the subjects walked shorter distances in the presence of an observer compared to the distance walked at the beginning of the study, suggesting they became partially habituated. However, they still walked further distances compared to individuals in the absence of observers, indicating that the subjects were not fully habituated yet. This is in line with our model results that advocates that an average of 454 h of human presence per individual (in this specific group of giraffes and under the same circumstances) would be required for a full habituation, and human presence after finishing this study was on average 333 h per individual (Table 1). Of course, the hours an observer would have to be present to habituate giraffes may differ at different study sites, as it is related to stressor intensity [17] and environmental specificities (e.g., human observers for research, tourism, intensive management, and human settlements). In addition, habituation success can be influenced by environmental experience in the past [43]. At RNR, hunting activities took place until the month before the study started. Even though there was no hunting during the study, this likely influenced the habituation process. Bond et al. (2020) [9] suggests that giraffes in Tanzania are intolerant of people approaching them on foot due to poaching, but they appear to be tolerant of tourist vehicles. In addition, there are no large predators at RNR, and giraffes who are used to large predators might react differently to the presence of human observers. For example, Nowak et al. (2014) [1] showed that observers formed a “human shield” for samango monkeys, as terrestrial predators stayed away from human observers and indirectly protected the monkeys. The monkeys altered their behavior and fed more at ground level in the presence of humans compared to when humans were not present. The habituation process might be hasted if predators stay away from giraffes due to human presence. Bond et al. (2019) [11] found that female giraffes with young calves were more frequently closer to traditional homesteads (bomas) compared to individuals without young calves, indicating that predator behavior is disrupted near bomas. 

If an animal alters its activity budget and spends more time walking due to human presence, as shown in this study, this may prevent them from resting and feeding [44]. This could affect the reproductive success of the individuals [45], especially for species that require high energy for breeding. Male giraffes’ reproductive tactics involve a trade-off between food intake and opportunities for breeding, which is a high-energy investment [23,24,25]. In addition, a recent study shows that giraffe bulls physically fight over a female in estrous [46]. For the blue-black grassquit (*Volatinia jacarina*), there is a relation between body condition and winning a male–male competition [47]; therefore, alteration of behaviors with the consequence of lower energy intake can result in not getting access to a female. Furthermore, giraffe cows conceive while lactating, and pregnancy and milk production both require a lot of energy [26]. Failure to maintain their energy balance can induce reproductive costs such as a decline in survival, lower probability of future reproduction, and reduced offspring’s fitness [48].

This study shows that unhabituated individuals present in the herd increases the distance walked by the focal individuals with human presence. This indicates that if unhabituated giraffes walk away from human observers, and habituated giraffes do not necessarily do that, then human observers can reduce opportunities for habituated giraffes to interact with unhabituated animals, which means that human presence can temporarily influence the giraffe’s social structure. Mountain gazelles (*Gazella gazella*) in Israel stayed in bigger groups with low human disturbance levels and in smaller groups with high disturbance levels [3]. Further, a recent study on giraffes shows that females inhabiting areas closer to human settlements in northern Tanzania displayed weaker social bonds compared to giraffes who lived further away [9]. It is known that social bonds are also important for reproductive success [49], as shown in feral horses [50], African elephants [51], and baboons [52]. A social disruption can influence natural selection and the reproductive success of a species, and if anthropogenic influences affect giraffes’ social structures, it may also influence their reproductive success. 

Wildlife viewing of polar bears (*Ursus maritimus*) in Canada influenced the bear’s vigilance behavior [10]. Polar bears increased their frequency of head-ups during resting in the presence of tundra vehicles; however over time, habituation to the vehicles occurred [10]. Jack et al. (2008) [18] suggests that habituated *Cebus capucinus* showed less observer-directed responses compared to non-habituated groups, but still responded to human observers even after 20 years of habituation. In this study, the giraffes did not spend less time OO at the end of the study compared to the beginning. This may indicate that they did not become habituated regarding their vigilance. However, our personal observations indicated that towards the end of the study, the giraffes more frequently OO out of curiosity and sometimes walked up to the observers, OO and ruminating, whereas, in the beginning, they often walked away while OO or stood up straight with an s-curved neck (a posture often seen when an animal is vigilant). Because curiosity is an objective observation, we did not differentiate between OO out of curiosity and otherwise. Van Krunkelsven et al. (1999) [53] showed that curiosity is part of the process of habituation of bonobos (*Pan paniscus*): individuals, at a stage, showed curiosity and as they became more habituated, showed an increase in ‘ignore’ reactions. This may indicate that, in our case, the giraffes did become habituated to human presence over time (the reason for altered OO over time) but were not fully habituated. Nevertheless, whether they showed OO out of curiosity or for other reasons, human observers did have an impact on giraffes’ behavior. 

As for the distance walked, the percentage of time spent OO of the focal individuals increased in the presence of unhabituated individuals. This advocates that it is best to habituate a whole population rather than only the study subjects, so to limit the effect of intraspecific (flight and vigilance) imitation responses. However, habituating whole populations can be challenging, as giraffes live in fission–fusion dynamics with large home ranges. Intraspecific responses to distress calls and warning signs are well known amongst several species [54,55,56]. Interspecific mutualism where species respond to another species’ alarm call is also common [57]. Schmitt et al. (2016) [58] showed that zebras (*Equus quagga*) reduced vigilance in the presence of giraffes, indicating that they receive cues from giraffes on predation risk. Based on personal observations, other species that were not habituated influenced giraffes’ movement and behavior when human observers were present. Giraffes responded to unhabituated species who were vigilant or ran away from human observers, like zebra, blue wildebeest (*Connochaetes taurinus*), and eland (*Taurotragus oryx*). When another species reacted to the human observers, the giraffes would often either run as well or become vigilant for a while. 

Inter-individual variation in behavior towards habituation was observed in three-spine sticklebacks (*Casterosreus aculeatus*); especially, aggression responses to different stimuli varied amongst individuals [6]. In line with that study, our results showed individual differences for giraffes: some subjects were significantly less inclined to spend time on OO than others. Since all target subjects were born at RNR and had similar human contact over the years, it indicates that there might be a personality difference in vigilance amongst the seven target giraffes. 

Grouping and vigilance amongst prey species are complementary responses to risk [59]. For example, the vigilance behavior of impalas (*Aepyceros melampus*), wildebeest (*Connochaetes taurinus*), and springbok (*Antidorcas marsupialis*) decreased with an increase of herd size [60,61]. This is consistent with the results of this study; herd size and number of human observers influenced the percentage of time spent OO. The number of human observers is especially important to consider, as the ecotourism sector is fast growing [62], and giraffes are one of the most important species of interest to international visitors in South Africa [63]. Buckley et al. (2016) [64] stated that many species rely on ecotourism for conservation funds, but concurrently, may also suffer from the ecological impacts of ecotourism. Human–wildlife encounters involve a complexity of interaction stimuli with problematic habituation consequences; hence, habituation must be treated critically when considering sustainable tourism management [65]. Wildlife viewing requires concentrated and well-defined locations, and interactions with wildlife must be constant and predictable [66]. 

In neurobiology, the term “habituation” has been used in the literature for several decades [67,68] and refers to the reduction of physiological responses caused by a repeated stressor [69]. Consistent with data for other species (e.g., rats [17], *Cebus capucinus* [18], western lowland gorillas [19], little bustards [7]), our results showed that the presence of human observers had a slight effect on fGCM concentration. Our model suggests an expected time to habituation of 427 h for that specific group of giraffes, under the same circumstances. This is in line with the 454 h suggested by our model on distances walked. Giraffes undergoing habituation or giraffes living in highly populated tourist places may perceive related stressors over a prolonged period and therefore could potentially have lower immune and reproductive functions [16,27,28,29,70,71]. 

The results of this study add to the growing literature suggesting that anthropogenic influences affect the behavior and, potentially, the social structures of giraffes [9,72] and highlight the importance of considering the effects that anthropogenic activities have on an animals’ wellbeing. 

## 5. Conclusions

Given that giraffes alter their behavior in the presence of humans, with possible effects on their social structure and/or interactions and related fGCM levels, it is important to evaluate the degree of human impact on giraffe populations for conservation decisions. Undergoing habituation to reduce behavioral and physiological changes is therefore an imperative process for giraffe conservation in areas with high anthropogenic impact. However, further research is necessary to test if the preliminary findings of this study also apply to other scenarios. Anthropogenic influences may have greater effects in, for example, e crowded tourist places.

## Figures and Tables

**Figure 1 animals-11-01239-f001:**
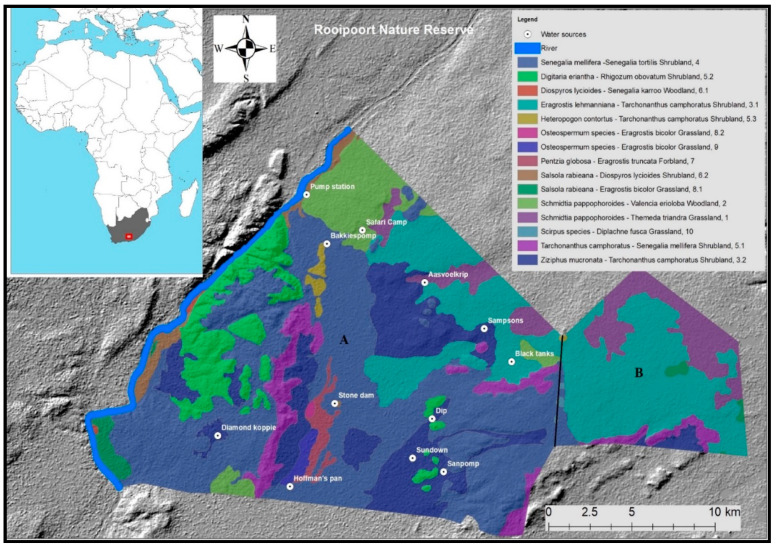
Study site (area A) at Rooipoort Nature Reserve including a vegetation map (identified by Bezuidenhout, 2009) and permanent waterhole locations on a digital elevation topography map. Area B is fenced off and inaccessible for giraffes.

**Figure 2 animals-11-01239-f002:**
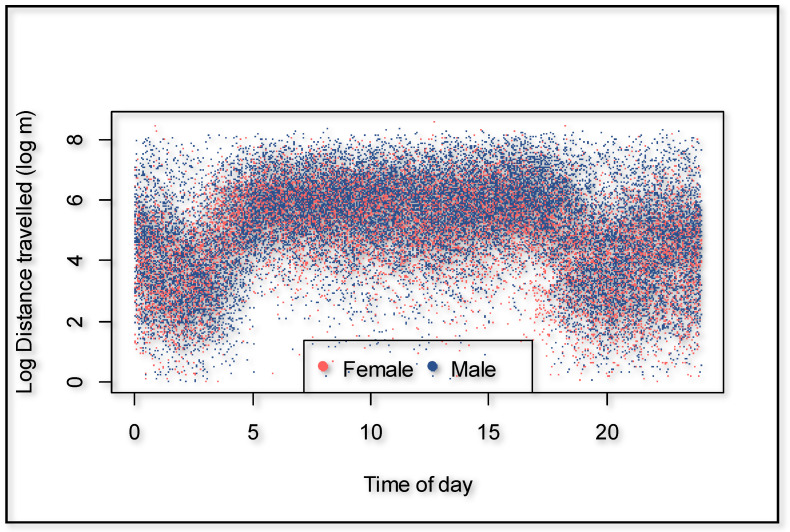
Distances traveled for males and females per hour over 24 h.

**Figure 3 animals-11-01239-f003:**
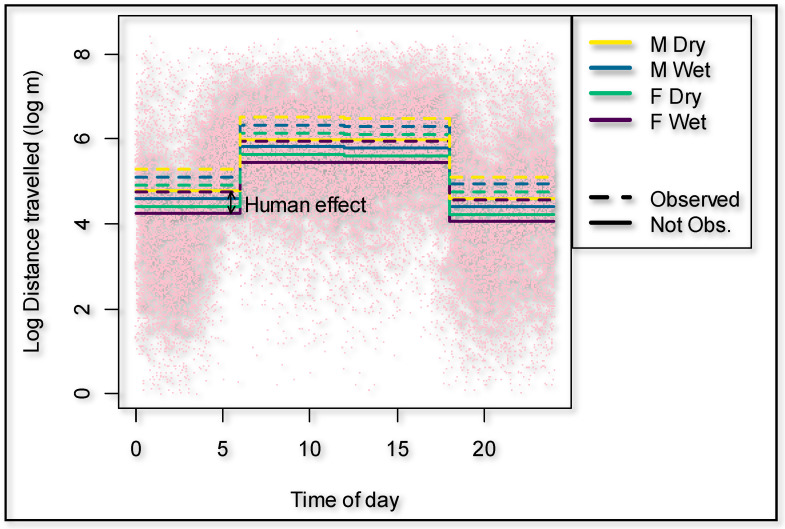
Effect on the distances traveled by giraffes depending on human observers present vs. not present as well as season and time of the day.

**Figure 4 animals-11-01239-f004:**
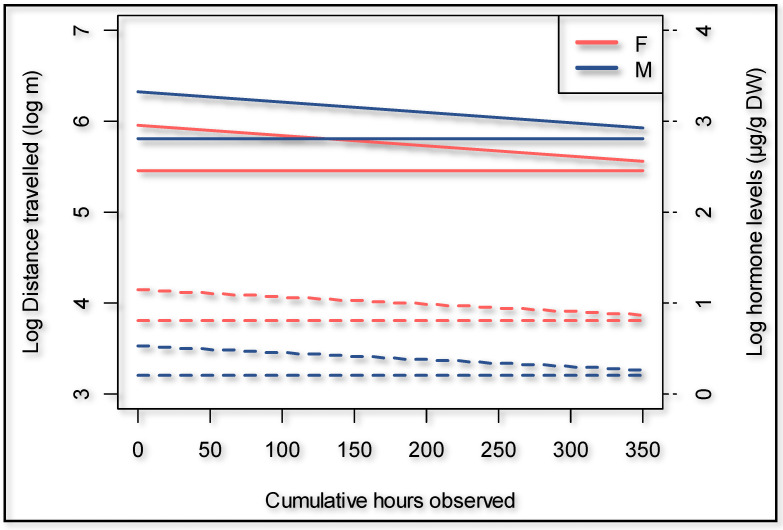
Giraffe habituation shown as distance walked as well as fecal glucocorticoid metabolite concentrations over time. Dotted lines are the hormone levels, straight lines are the distances travelled.

**Figure 5 animals-11-01239-f005:**
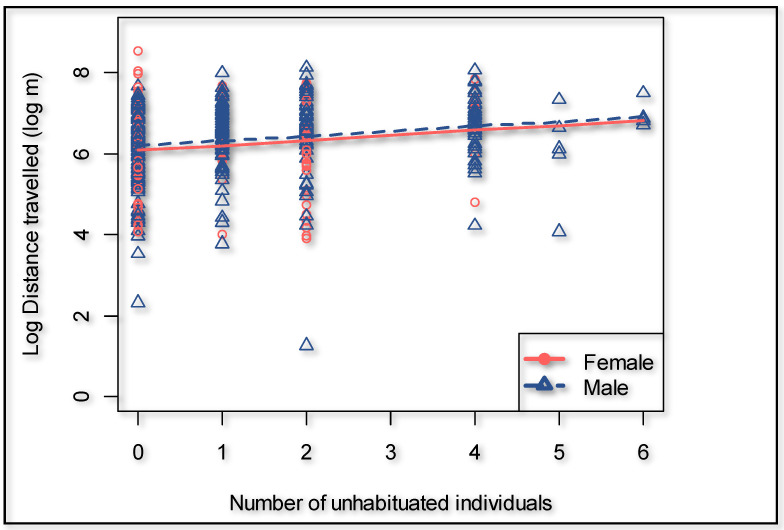
Effect of the presence of unhabituated giraffes on the distance traveled by the focal individuals when human observers are present.

**Figure 6 animals-11-01239-f006:**
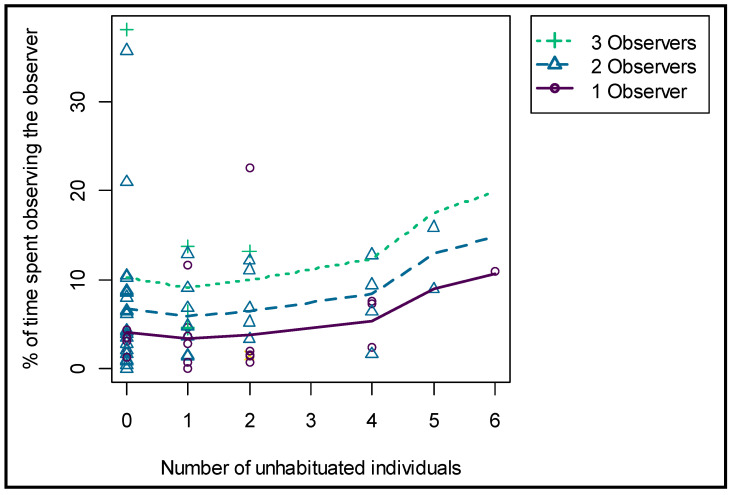
Effect of the number of observers and the number of unhabituated giraffes present on the time spent OO by the focal individuals.

**Figure 7 animals-11-01239-f007:**
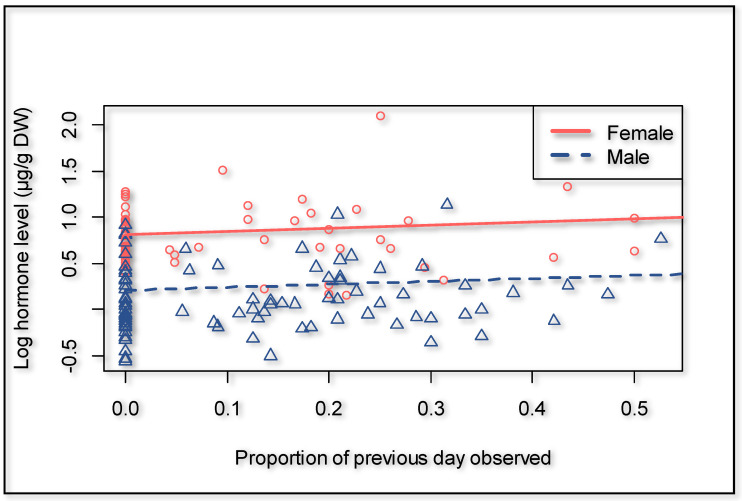
Effect of the presence of human observers on fGCM concentration, showing the proportion with respect to the previous day observed, to account for the time lag in hormone metabolite expression (0.0 = observer absent).

**Table 1 animals-11-01239-t001:** Number of logs, hours observed, hours of human presence per individual throughout the study period.

ID ^1^	NLogs	Hours Observed	Hours Present
F2	5865	339	389
F6	3675	280	344
F7	2605	150	188
M1	6059	324	395
M4	6543	255	328
M5	4318	284	367
M7	4707	229	317
33 772	33,772	1861	2328

^1^ F = female and M = male.

**Table 2 animals-11-01239-t002:** Average distance traveled (m) per hour per individual.

Males				Females			
ID	Average Distance (m)	SEM	SD	ID	Average Distance (m)	SEM	SD
M1	385	6.616	515	F2	278	5.198	398
M4	393	6.263	507	F6	286	5.804	352
M5	455	9.211	605	F7	255	7.076	361
M7	400	7.467	512				
Total average	408			Total average	273		

## Data Availability

The data that support the findings of this study are openly available in ufs.figshare at https://doi.org/10.38140/ufs.14221889 “16 March 2021”.

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
