# Peer review of "Anthropogenic Influences on Distance Traveled and Vigilance Behavior and Stress-Related Endocrine Correlates in Free-Roaming Giraffes"

_animals, 2021, doi:10.3390/ani11051239_

Round 1
Reviewer 1 Report
Comments on "Anthropogenic influences on distance traveled and vigilance behavior and its stress-related endocrine correlates in free-roaming giraffes"
by CPJ Scheijen , S van der Merwe , A Ganswindt , F Deacon
General comments
The authors report a complex, articulated study on vigilance, traveled distance and stress-linked hormonal levels in giraffes, with several aspects of of human influences on these specie.
In my opinion, the paper seems sound, interesting and complete, taking into account the difficulties encountered when free-range wild species are studied.
The positive aspects of the research include for example the use of GPS, the immediate (<30') collection of feces, and the punctual use of linear models, balancing the introduction of the single factors.
Besides little 'adjustments' on the body of the manuscript (see specific comments), I warmly recommend the acceptance of the paper in Animals.
Specific comments
- Please report the species names in italic
-Maybe the acronym AIC stands for 'Akaike information criterion': please specify the meaning of the acronym
- Please include the name of statistical software used
- Table 1: please insert dispersion parameters for data (SD or SEM)
- Figures 2, 3, 4, 5: please insert in y axis the units for travelled distance
- What kind of metabolites include fGCM? It would be possible to include the measure units for fGCM in the text and graphs?
Reviewer 2 Report
Anthropogenic influences on distance traveled and vigilance behavior and its stress-related endocrine correlates in free-roaming giraffes.
Using interdisciplinary research methods, the authors provide new insight into how anthropogenic activities may influence the vigilance behavior, social structure, and stress-related hormones in giraffes. I found this study very interesting, particularly given that anthropogenic impact in giraffes has only received limited attention, yet this is an important area of study, and indispensable to set conservation and welfare actions. The manuscript is easy to follow, very well written, and the methods and analyses are both rigorous and well-executed. The results are accurately interpreted and I believe that this paper will make an important contribution to the field of giraffe behavior and ecology. I would like to congratulate the authors on an excellent study. I only have minor comments and changes to be considered:
Line 70
Typo in the author's name. The correct spelling is “Canine”. Also, please add reference number 12.
Line 99
Font size of “Cebus imitator” needs to be adapted.
Line 180
Please add a comma between “hours observed” and “hours of human presence..”
Line 160
Comment: The authors write that hunting still occurred at the reserve shortly before commencing the study, and I strongly assume that it was done using vehicles and not by foot. Further, I could imagine that the event of being chased and eventually being captured might leave a “long-term” negative association with vehicles/helicopters, not only in the focal animal but collectively, also in other group members witnessing these events.
Could it be that the giraffes were rather vigilant towards the observer vehicles and not the observer itself?
Line 222
I think “spot pattern” is the more appropriate term than “skin pattern”? after all their pattern results from their short, dense fur.
Referring to my previous comment for Line 160, I would kindly ask to provide some more information about the behavioral observations. Did the author choose only certain behaviors from Seeber et al 2012 for their focal sampling protocol? If yes, a supplemental table might be useful for the reader.
Further, to me it is not clear how observations were conducted. Did the authors and observer use vehicles for their observations? If yes, did each observer its own vehicle, meaning did the number of vehicles increased with the number of observers?
If 3 observers were sharing one vehicle for their observations, I think this would have consequences for the analysis, as the vigilance would be directed towards the vehicle and not the 3 observers inside?
Line 233
Fecal samples from giraffes not exposed to observer presence were collected using the time lag. Fecal samples associated with observer presence were collected with 30 minutes of defecation.
Perhaps the authors could provide information about how they managed the right timing for freezing feces within the standardized 30 minutes time window when no observers were present? Maybe I misunderstood, but does it mean that the observer followed that last GPS data point and took fecal samples only when no giraffe was in the vicinity?
Line 255
Without intending to disabuse the authors… Although knowledge on giraffe nocturnal activity behaviors in the wild was very limited, including nightly antipredator strategies or vigilance behavior within a group, recent studies suggest that giraffes are not strictly diurnal, and should be considered cathemeral. For example, it was shown that giraffes exhibit a guarding system during nocturnal phases of increased predation risk; while some herd members remain in vulnerable sleep postures, others are vigilant while feeding or standing next to sleeping conspecifics. Further giraffes have very good vision in mesopic light and have been observed to feed and move during nighttime as well, particularly on moonlit nights.
E.g. https://onlinelibrary.wiley.com/doi/full/10.1002/ece3.6106,
https://doi.org/10.1016/j.beproc.2020.104178
Line 287-289
Please remove this paragraph as it is part of the author guidelines
Line 415
Please use italic format for ‘Ursus maritimus’
Line 418
Please use italic format for ‘Cebus capucinus’
Line 429
Please use italic format for ‘Pan paniscus’
Line 441
Reference 54-56 refers to bird acoustic studies only.
This might be a good time to add the finding that zebras eavesdrop on giraffes to obtain information about predation risk.https://doi.org/10.1093/beheco/arw015
Line 454
Please add the word "might" and tone down: “..it indicates that there might be a personality difference.."
